# Tree growth is better explained by absorptive fine root traits than by transport fine root traits
Anvar Sanaei [1] ✉, Fons van der Plas [1,2], Hongmei Chen [1,3], Sophie Davids[1], Susanne Eckhardt[1], Justus Hennecke[1,4], Anja Kahl [1], Yasmin Möller[1], Ronny Richter [1,4], Jana Schütze[1], Christian Wirth[1,4,5] & Alexandra Weigelt [1,4]

Although the interest in root traits has increased in recent years, we still have limited knowledge of (i) whether functionally different fine roots—absorptive versus transport roots—have similar trait coordination and (ii) how they help to explain plant performance, such as growth. We measured traits of 25 European broadleaved tree species growing in a research arboretum to study (i) the coordination of root traits within absorptive and transport fine roots and (ii) the degree of trait-tree growth relationships. To do so, we combined a suite of morphological and anatomical traits for each of the absorptive and transport roots. Despite remarkable differences in average trait values between absorptive and transport roots, our study shows that trait coordination within absorptive and transport roots is relatively similar. Our results also show that, for the selected traits, tree growth is better explained by absorptive root traits than by transport root traits and is higher in species with thinner roots. The stronger relationship between absorptive roots and tree growth highlights that roots mostly involved with resource absorption are more important in explaining tree growth than transport roots, which are mainly responsible for resource transportation.

Functional traits of plants are used to comprehend plant community structure, assembly and functioning[1,2]. Plant traits reflect different plant strategies and control how plants respond to the environment[1,3]; hence, they have the promise to answer how and why plant performance differs among species[4]. A suite of associated plant traits known as the leaf economics spectrum (LES) has been established at the leaf level[5,6]. The LES defines a functional gradient from leaves with conservative resource use to those with an acquisitive strategy, the latter providing a fast return on investment, thus being associated with high growth rates[6]. According to this rationale, species with acquisitive strategies are characterized by high leaf nitrogen content but lower leaf toughness (LT), leaf mass per area and leaf dry matter content (LDMC), while species with conservative strategies are characterized by the opposing leaf traits. The success of the LES in elucidating variations in leaf traits and predicting plant performance has stimulated researchers to expand the economic theory to fine roots, proposing a two-dimensional space of roots known as the root economics space (RES)[7]. The first dimension is known as the collaboration gradient that ranges from species with a high root diameter (RD) offering space for arbuscular mycorrhizal fungi (AMF)

to species with a greater specific root length (SRL). The second RES dimension, known as the conservation gradient, is equivalent to the classical LES with high root nitrogen content representing a fast-growth strategy and high root tissue density (RTD) representing a slow-growth strategy.

Many ecological studies on root traits define fine roots based on an arbitrary diameter size, and often implicitly assume roots within this size class to be homogenous in their functioning[8]. However, plant species typically possess hierarchical root systems. As a result, fine roots are composed of a collection of very heterogeneous orders and branches differing in morphology, architecture, anatomy, and longevity[8–11] as well as in microbial associations[12,13] and thus different root orders perform different functions[11,14]. Through this understanding, fine roots have been classified into two distinct groups based on their functional roles. The first group, absorptive roots (order ≤ 3), is responsible for resource uptake and serves as a hotspot for biotic interactions with soil microbes and mycorrhiza[11,15]. The second group, transport roots (order > 3), is most important for transport and storage[11,15]. Thus, the capacity of resource transportation increases while absorption capacity decreases with increasing root order[11]. However, the

[1]Institute of Biology, Leipzig University, Leipzig, Germany. [2]Plant Ecology and Nature Conservation Group, Wageningen University, Wageningen, The Netherlands. [3]Lancaster Environment Centre, Lancaster University, Lancaster, UK. [4]German Centre for Integrative Biodiversity Research (iDiv) Halle-Jena-Leipzig, Leipzig, Germany. [5]Max-Planck-Institute for Biogeochemistry, Jena, Germany. ✉e-mail: anvar.sanaei@uni-leipzig.de

transition from absorptive to transport is species-specific and might occur gradually, causing variation in the transitional root order among plant species[11,16]. Moreover, the lifespan and root diameter of root segments depend on the root order, and consistently increase from the distal to the proximal end[8,17]. Given this, absorptive roots located at the distal end have a smaller diameter and greater SRL compared to transport roots, and exhibit a shorter lifespan[17,18]. On the other hand, transport roots, characterized by a larger diameter and longer lifespan, emerge later in the developmental process as a consequence of secondary growth, resulting in greater RTD and lower SRL[17,18]. In addition, in a root system, anatomical changes across root orders occur mainly due to shifts in physiological functions from resource uptake to transport and storage[10,19]. As such, a higher percentage of cortex area, which is characteristic of absorptive roots, is considered an indication of the capacity for resource absorption and mycorrhizal colonization[20–22]. Conversely, a higher stele diameter is known as an indicator of resource transportation in transport roots[10,21,23]. Generally, a key feature of higher root orders is that the proportion of the cortex decreases or disappears, while the stele expands due to secondary growth[10,24,25]. In this regard, it has been reported that the cortex expanded more rapidly than the stele in the first root orders, while higher root orders showed the opposite pattern[21]. Despite this heterogeneity in absorptive and transport root traits, the relative importance of absorptive and transport roots for ecosystem functions such as tree growth is still unexplored.

Forest ecosystem functioning directly and indirectly depends on variation in plant functional traits[26,27]; thus, studying the link between plant functional traits and ecosystem functioning is important for a mechanistic understanding of forest functioning[28,29]. Consequently, there has been a lot of interest in identifying the relationship between leaf functional traits and forest functioning[4,26,27]. For instance, along with the LES, tree annual growth was positively related to acquisitive traits, characterized by a high specific leaf area (SLA) and stomatal density in subtropical forests[30], and a high leaf nitrogen content and SLA in temperate forests[31]. In principle, such relationships have been attributed to higher photosynthetic capacity and a higher potential for a quick return on investment of resources in fast-growing species, leading to a higher growth rate[5,6]. Even though linking functional traits and plant performance is important, the majority of the studies have reported rather weak associations between plant functional traits and plant performance. For example, only 3.1% of variance in tree growth was explained by leaf traits at the global scale in forests[26] and 4.8% of variance across functions by leaf and root traits together in grasslands[32]. The reasons for such weak links could be due to the use of species-level mean trait data rather than individual-level trait data and/or using single traits rather than multiple traits, thereby weakening the strength of the relationships between plant functional traits and plant performance. The former might be attributed to the fact that different individuals of the same species respond differently to environmental variables[33]; for example, there is some evidence that individual-level trait data improves the degree of trait-growth relationships[34,35]. Besides, the performance of trees is contingent upon the synchronized functioning of leaves and roots, where the role of leaf functional traits in photosynthesis is well established[6,28]. However, fine roots serve a variety of functions, such as acquiring resources and interacting with soil organisms, all of which influence plant performance[11,36–38]. Yet, our understanding of the relative importance of fine root traits for tree growth significantly lags behind that of leaf traits, partly due to the difficulty of sampling and/or measuring root traits[37]. A few recent studies have examined the explanatory power of root traits—in combination with leaf traits—on tree growth, in which, for fine roots they focused only on absorptive roots (the first two or the first three root orders)[31,39,40]. Recently, Shen et al.[39] showed that acquisitive leaf traits had a higher explanatory power than fine root traits for relative growth rates for height across tree species, even though SRL and RTD were significantly correlated with the relative growth rates for height of individuals. By contrast, Da et al.[31] found that the conservation gradient of absorptive root traits explained forest aboveground carbon storage and woody biomass productivity better than conservation gradients in leaves and absorptive root collaboration gradients. Although few existing

studies have been restricted to the effects of absorptive fine root traits on tree growth, the simultaneous effects of absorptive and transport root traits have so far been unexplored. Besides, little is known about the effects of anatomical root traits on tree growth. Altogether, this highlights the necessity of examining the trait coordination within functionally discrete fine roots—absorptive and transport roots—as well as examining their relative importance for tree growth, either with or without the combination of leaf traits.

By using 25 European broadleaved tree species growing in a research arboretum in Germany (Table 1), this study aims to quantify the coordination within absorptive and transport fine roots and determine their explanatory power for tree growth, either with or without the combination of leaf traits. More specifically, this study tests the following three hypotheses: First, due to differences in the morphology and anatomical structures between absorptive and transport roots[8,10,11], we hypothesized (H1) that absorptive and transport roots do not necessarily demonstrate similar trait coordination patterns. Second, given the crucial functions of absorptive and transport roots in below-ground processes and functioning[11,12], we hypothesized (H2) that both absorptive and transport root traits are important for tree growth. Third, considering that tree growth relies on concurrent acquisition of above- and below-ground resources, which can be provided through both leaves and roots[5,36], we hypothesized (H3) that tree growth is better explained by a combination of all organs directly responsible for resource acquisition, that is, leaves and absorptive roots, as compared to using root or leaf traits alone.

## Results
### Covariation in absorptive and transport root traits
We found that root traits, except arbuscular mycorrhizal colonization rate (MCR), substantially differed between different fine root types. In particular, compared to transport roots, absorptive roots showed higher specific root length (SRL) and cortex to stele ratio (C:S), while transport roots had higher root diameter (RD) and RTD (Fig. 1). Furthermore, pairwise trait correlations in absorptive roots revealed that there were much more negative relationships between RD with both RTD and LDMC as well as between MCR and SRL compared to transport roots (Fig. 2). Although there was a much stronger positive correlation in absorptive roots between RD with both MCR and C:S compared to transport roots (Fig. 2).

The PCA of absorptive root traits showed that the first two axes together captured 78% of the variability (Fig. 3a, Table S1). The first principal component (PC1) axis is positively associated with RD and MCR, and the second principal component (PC2) axis is positively and negatively related to SRL and RTD, respectively (Fig. 3a, Table S1). The first two PC axes of transport root traits together explained 67% of the variability (Fig. 3b, Table S1). The PC1 of the transport root traits was also positively associated with RD, but unlike in absorptive root traits, it was in addition negatively associated with SRL (Fig. 3b, Table S1). The PC2 was positively related to MCR and C:S and RTD was negatively loaded on the third PC axis (Fig. 3b, Table S1). Considering only leaf traits, the PCA showed that the first two axes together captured 88% of leaf trait variation (Fig. 3c). PC1 was negatively associated with LDMC and LMA, and PC2 was negatively associated with LT (Fig. 3c, Table S1). The results of the PCA based on the whole set of absorptive root and leaf traits showed that the first two axes accounted for 62% of variation (Fig. 3d, Table S1): PC1 was positively related to the RD and MCR while negatively related to LDMC and PC2 was mainly positively and negatively related to SRL and RTD, respectively (Fig. 3d, Table S1). The results based on the whole set of transport root and leaf traits showed that PC1 and PC2 accounted for 52% of variation: PC1 was positively related to MCR, while it was negatively associated with LDMC and LMA (Fig. 3e, Table S1). PC2 was positively associated with SRL, while being negatively related to RD (Fig. 3e, Table S1; note that here both PC axes are flipped). While the overall trait coordination of absorptive and transport roots is relatively similar, MCR and C:S decoupled from RD in transport roots, resulting in MCR and C:S shifting to the second PCA axis in transport roots (Fig. 3b, S2). The PCA results of combining both absorptive and transport roots showed that root traits in absorptive roots were strongly related to the

**Table 1 | List of 25 tree species used in this study**

| Species | Abbreviations | Family | Order |
|---|---|---|---|
| *Acer campestre* L. | Ace cam | Sapindaceae | Sapindales |
| *Acer platanoides* L. | Ace pla | Sapindaceae | Sapindales |
| *Aesculus hippocastanum* L. | Aes hip | Sapindaceae | Sapindales |
| *Alnus glutinosa* Medik. | Aln glu | Betulaceae | Fagales |
| *Alnus incana* L. | Aln inc | Betulaceae | Fagales |
| *Betula pubescens* | Bet pub | Betulaceae | Fagales |
| *Castanea sativa* Mill. | Cas sat | Fagaceae | Fabales |
| *Corylus avellana* L. | Cor ave | Corylaceae | Fagales |
| *Euonymus europaeus* L. | Euo eur | Celastraceae | Celastrales |
| *Fagus sylvatica* L. | Fag syl | Fagaceae | Fagales |
| *Frangula alnus* L. | Fra aln | Rhamnaceae | Rosales |
| *Fraxinus excelsior* L. | Fra exc | Oleaceae | Lamiales |
| *Fraxinus ornus* L. | Fra orn | Oleaceae | Lamiales |
| *Juglans nigra* L. | Jug nig | Juglandaceae | Fagales |
| *Mespilus germanica* L. | Mes ger | Rosaceae | Rosales |
| *Ostrya carpinifolia* Scop. | Ost car | Corylaceae | Fagales |
| *Prunus mahaleb* L. | Pru mah | Rosaceae | Rosales |
| *Quercus cerris* L. | Que cer | Fagaceae | Fagales |
| *Quercus robur* L. | Que rob | Fagaceae | Fagales |
| *Quercus rubra* L. | Que rub | Fagaceae | Fagales |
| *Salix alba* L. | Sal alb | Salicaceae | Malpighiales |
| *Salix pentandra* L. | Sal pen | Salicaceae | Malpighiales |
| *Sorbus aucuparia* | Sor auc | Rosaceae | Rosales |
| *Sorbus torminalis* (L.) Crantz | Sor tor | Rosaceae | Rosales |
| *Ulmus laevis* Pall. | Ulm lae | Ulmaceae | Rosales |

corresponding traits in transport roots (Fig. S2a). Specifically, RTD of both root types was positively associated with PC2, MCR and C:S to PC1, and SRL negatively to the PC2 axis. However, the exception was RD, which mostly was related to PC3 for transport roots and to PC1 for absorptive roots (Fig. S2a, Table S2).

## The relationships between fine root and leaf traits and tree growth

Our results of linear regressions between PC1 and average basal tree area increment reveal that absorptive root traits were negatively associated with tree growth ($R^2 = 0.35$, $P < 0.01$; Fig. 4a), showing a higher growth for trees with thinner absorptive roots (lower RD) and lower MCR, while there was no significant relationship between tree growth and PC2 (Fig. 4a). In contrast, while there was no significant relationship between tree growth and PC1 of transport root traits (Fig. 4b), PC2 revealed a significant relationship with tree growth ($R^2 = 0.16$, $P < 0.05$; Fig. 4b), showing a higher growth for trees with lower C:S and MCR. The linear regressions between PC1 of leaf traits and average basal tree area increment showed that leaf traits were related to tree growth ($R^2 = 0.19$, $P < 0.05$; Fig. 4c), showing a higher growth for trees with higher LDMC and LMA. Moreover, PC1 of absorptive root and leaf traits together explained even more variance in tree growth ($R^2 = 0.39$, $P < 0.001$; Fig. 4d), where trees with higher RD and MCR but with lower LDMC showed lower growth (Fig. 4d; Table S1). Finally, PC1 of a combination of transport root traits and leaf traits also revealed a significant relationship with tree growth ($R^2 = 0.18$, $P < 0.05$; Fig. 4e), where trees with higher MCR but with lower LDMC and LMA showed lower growth (Fig. 4e; Table S1). The explanatory power of absorptive root and leaf traits on tree growth was much stronger (Fig. 4d) than that of transport root and leaf traits (Fig. 4e). In line with the PCA results on multiple traits, single root and leaf traits were also correlated with tree growth (Fig. S3). As such, absorptive root

traits (RD, MCR, and C:S) were significantly negatively correlated with tree growth ($P < 0.05$ to $P < 0.01$; Fig. S3a, d, e), while RTD was positively correlated with tree growth ($P < 0.05$; Fig. S3b). For transport roots, only MCR was significantly negatively associated with tree growth ($P < 0.05$; Fig. S3i). For leaves, LMA was significantly positively related to tree growth ($P < 0.05$), while LDMC was marginally positively correlated with tree growth (Fig. S3k, l).

Compared to using a single principal component (PC), the inclusion of additional PCs of absorptive roots, up to a cumulative explained variance of 70% (PC1:2), slightly improved the prediction of tree growth (Table 2, 5% increase). For transport roots, the multiple regression of using the first three PC axes (PC1:3) improved the explanatory power of tree growth estimation compared to using only the first PC (Table 2, 13% increase). However, using only one, two or three PCs did not change the explanatory power of tree growth for the combination of absorptive roots and leaf traits and leaf traits alone, respectively (Table 2). Although adding more PCs slightly enhanced the predictive power of traits for tree growth, absorptive roots still remained more influential than both transport roots and leaf traits. The summary of the models is provided in Table 2.

## Discussion

By functionally separating fine roots into absorptive and transport roots, we explored the coordination within absorptive and transport fine roots, which, based on our knowledge, has not been tested so far. Overall, we found that trait coordination within absorptive and transport roots is comparable. Specifically, mycorrhizal colonization, root diameter, and cortex-to-stele ratio were the key traits loading on the first PC axis. Furthermore, tree growth is better explained by absorptive than by transport root traits and was higher in species with thinner roots that were less colonized by arbuscular mycorrhizae, highlighting the role of efficient and independent exploration of soil resources.

### Covariation in absorptive and transport root traits

Despite significant differences between absorptive and transport root traits (Fig. 1), we found that, contrary to our first hypothesis (H1), coordination within absorptive and transport root traits was quite similar to each other (Figs. 3a, b, S2), with the exception of RD, which was decoupled from MCR and C:S in transport roots (Figs. 2, 3b). Our findings show that species with higher root diameter were highly related to mycorrhizal association, but this was true only for absorptive roots, similar to a previously published study[7]. In partial disagreement with our results, in another study different economic strategies were observed for thin (<247 μm) and thick (>247 μm) fine roots, where thin roots followed the resource acquisition-conservation strategy but thick roots did not[41]. It must be mentioned that[41] applied univariate regression analysis between root traits, not PCA for the trait coordination. The specific fine root diameter cutoff, limited number of species, and/or inclusion of root nitrogen concentration, which we did not measure, can contribute to the different observed patterns. This again highlights the importance of trait selection for the outcome of studies on trait coordination patterns[42].

Against our expectation, there was no significant difference in mycorrhizal colonization between absorptive and transport roots (Fig. 1), which is contrary to the generally acknowledged notion that higher root orders (or transport roots) are not or less colonized by mycorrhizae[11,12]. Indeed, transport roots possess lower potential for mycorrhizal colonization due to their thinner cortex (or presence of periderm), providing smaller space for mycorrhizal colonization[11,22,43]. One reason for our inconsistent results is that species-specific differences in mycorrhizal dependence might affect the overall colonization of the roots with mycorrhizae[21]. There is some evidence that, for example, *Fraxinus rhynchophylla* Hance. has mycorrhizal colonization in fourth order roots and *Acacia auriculiformis* A.Cunn. ex Benth. is colonized even in fifth order roots, meaning that some species are more colonized by mycorrhizae than others even in higher root orders[10,24]. This is because plant species differ in the secondary growth development, and mycorrhizal colonization in higher root orders also confirms a higher dependency of those species on mycorrhizae for nutrient uptake[21]. This was

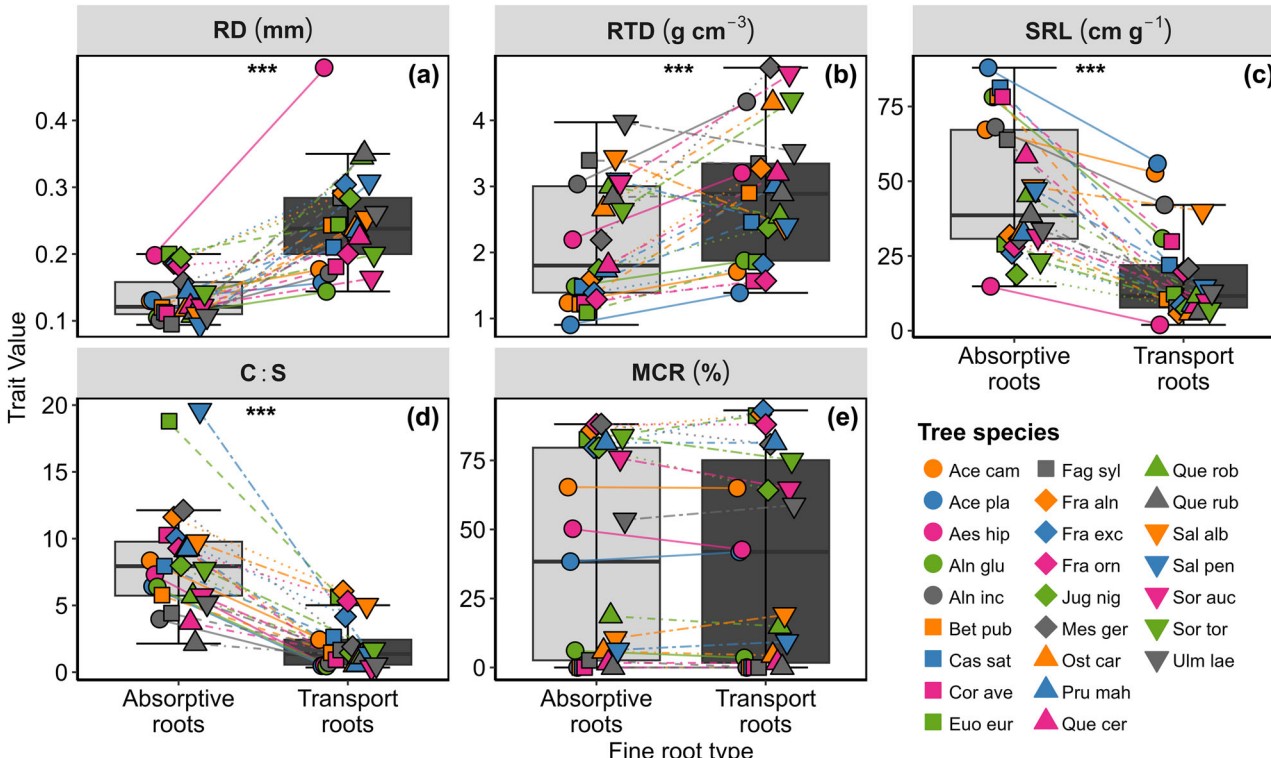

**Fig. 1 | Mean root trait values for absorptive and transport roots.** Changes in root traits between absorptive and transport roots. Absorptive roots are in light gray, while transport roots are in dark. Significant differences within fine root types are denoted by * ($P < 0.05$), ** ($P < 0.01$) and *** ($P < 0.001$). Lines illustrate how the traits of absorptive and transport roots vary across 25 different tree species.

Abbreviations: RD root diameter; RTD root tissue density; SRL specific root length; C:S cortex to stele ratio and MCR mycorrhizal colonization rate. Different point colors and shapes represent different tree species. See Table 1 for tree species name abbreviations. Data represented are the median (line in the middle) ± IQR.

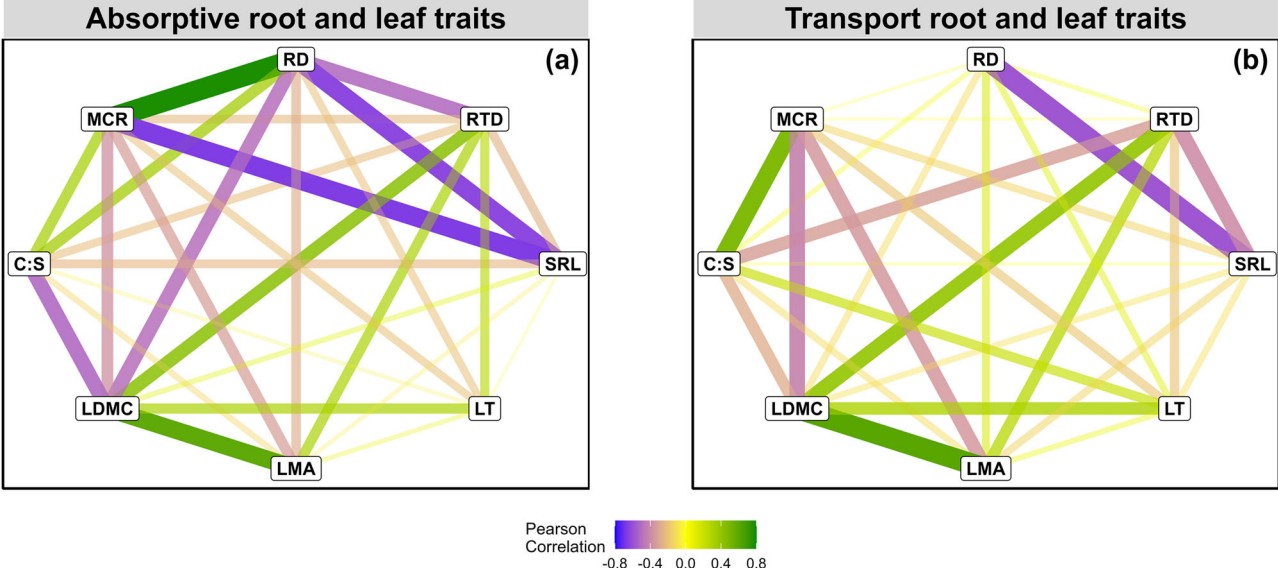

**Fig. 2 | Pairwise network trait correlations.** Pairwise trait correlations of **a** absorptive root and leaf traits and **b** transport root and leaf traits. Nodes represent root and leaf traits and line width represents the strength of the correlation. Green and blue lines represent positive and negative correlations, respectively. RTD root tissue density; SRL specific root length; RD root diameter; C:S cortex to stele ratio; MCR mycorrhizal colonization rate; LT leaf toughness; LMA leaf mass per area; and LDMC leaf dry matter content.

the case in our mycorrhizal colonization data. As such, order-based root mycorrhizal colonization data showed that for the majority of species, mycorrhizal colonization was greater in the lower root orders or remained on the same level in the higher root orders. Yet, in some species, like

*Fraxinus excelsior* L., *Euonymus europaeus* L. and *Salix alba* L. mycorrhizal colonization slightly increased with increasing root orders. Another possible explanation could be that the MCR varies with the age of tree species, with younger tree species potentially having a greater dependency than mature

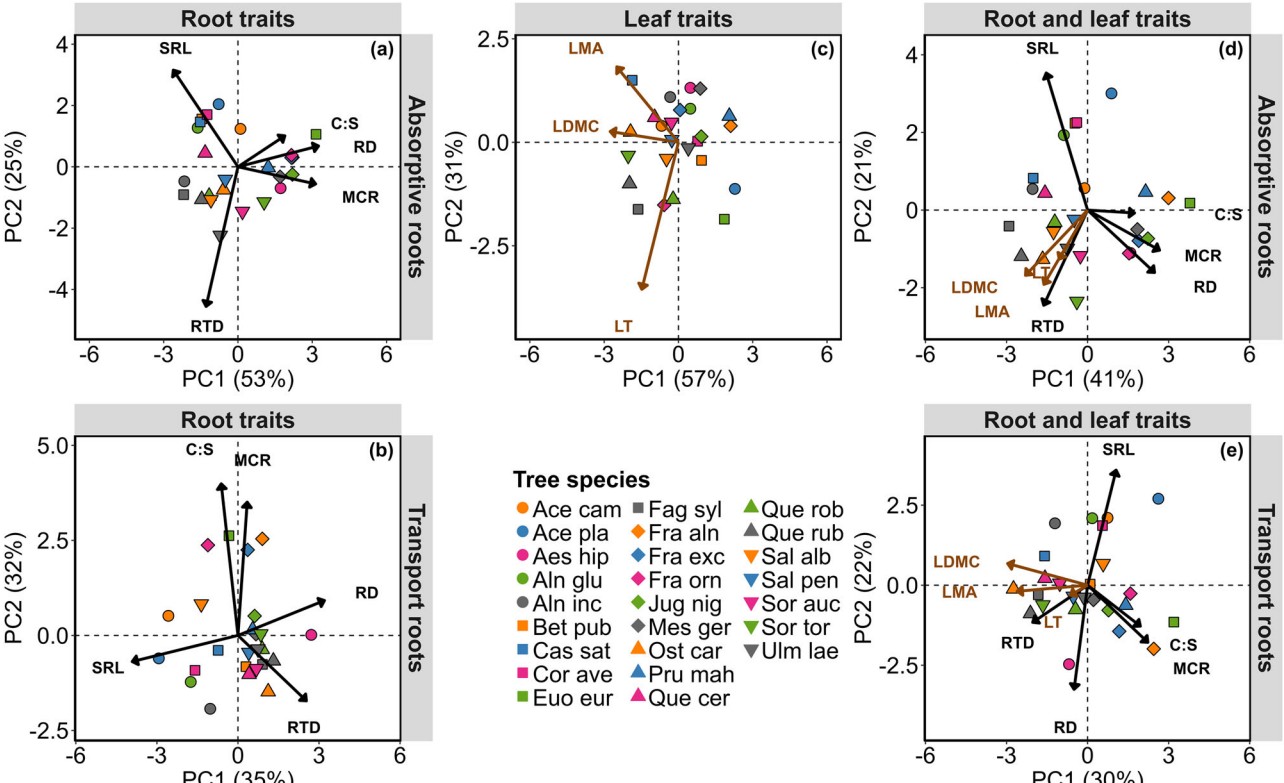

**Fig. 3 | Trait-trait relationships using principal component analyses.** Principal component analyses (PCAs) of species-levels of **a, b** absorptive and transport root traits, **c** leaf traits and **d, e** all root and leaf traits for both absorptive and transport root traits, respectively. The first and second PC axes of the whole set of transport roots and leaf traits (**e**) are flipped. RTD root tissue density; SRL specific root length; RD root diameter; C:S cortex to stele ratio; MCR mycorrhizal colonization rate; LT leaf toughness; LMA leaf mass per area; and LDMC leaf dry matter content. Different point colors and shapes represent different tree species. See Table 1 for tree species name abbreviations.

species[44]. This suggests a higher dependency of younger tree species on mycorrhizae for nutrient uptake, particularly during drought years. Given that the studied trees are relatively young compared to those trees in other studies[10,45], this might be one additional explanation for the higher colonization observed, even in higher root orders. While our overall results showed reduced mycorrhizal colonization in the fifth root order (Fig. S4), we argue that due to the high diversity of AMF and its appearance, the distinction of AMF structures in the samples is challenging, especially for 25 tree species across different root orders. First, fungal staining quality can differ between species. Second, the magnified intersection method[46] is based on counting presence/absence of mycorrhizal structures rather than density. Thus, one small hyphae counts the same as a root filled with hyphal structures, which could lead to biases in overall colonization density. Moreover, staining and counting do not indicate whether mycorrhizal structures are actively functioning and contributing to the root's nutrient acquisition processes. This method might be expanded by incorporating novel techniques such as root image analysis and molecular markers, which offer greater reliability and precision[47]. Altogether, this might lead to a bias in the overall differences in mycorrhizal colonization rates between absorptive and transport roots. Furthermore, due to the unique anatomical structures of different species and the resulting variation in root functions across root orders, further research is needed to precisely distinguish between the absorptive and transport functions of each tree species based on their anatomical changes.

By incorporating leaf traits into PCAs with absorptive and transport roots, trait coordination showed that conservative leaf traits were closely aligned with conservative root traits, reaffirming that the conservation gradients of both the LES and RES are correlated[6]. Similar results have been reported when leaf and root traits were pooled, indicating the same trade-offs between the fast–slow conservation gradient in root and leaf traits[48,49].

## Absorptive root traits better explain tree growth than transport root traits

Past attempts at exploring the contribution of fine root traits to plant performance have considered fine roots as a homogenous pool without regard to their distinct functional roles[32,50]. Thus far, our understanding of how fine roots contribute to tree growth stems from studies testing either the first two or three root orders[31,39,40], but there is no study testing the effects of functionally discrete fine roots on tree growth. By separating fine roots into absorptive and transport roots, we found that absorptive fine root traits are highly correlated with tree growth, in partial support of our second hypothesis (H2). The greater contribution of absorptive root traits to tree growth compared to transport root traits can be attributed to the functioning role of absorptive roots within the plant system[11,15]. Within the plant, absorptive roots are mainly involved in soil-based resource acquisition (e.g., nutrients and water), which is directly linked to tree growth. More specifically, the absorptive root traits loaded on the PCA axis 1 (MCR, root diameter and C:S ratio) were the key traits associated with tree growth, highlighting the importance of thin roots with a 'do-it-yourself' strategy of resource uptake for tree growth[7,51]. Indeed, the positive associations among mycorrhizal colonization, root diameter and cortex-to-stele ratio are characteristic of absorptive roots[38], and we observed that those traits have stronger correlations in absorptive roots (Fig. 2). More precisely, our results showed that species with thicker roots that are more colonized by AMF[20,43] were negatively correlated with tree growth. Indeed, plants with thicker roots tend to have a longer lifespan and a smaller surface area, resulting in a smaller volume of below-ground resources explored and thus a high dependence on mycorrhizal colonization[8,52]. In contrast, SRL, as a part of the root collaboration gradient in RES, was positively correlated with tree growth, meaning that species with the ability to independently explore soil for resources have a higher growth rate. Similar results were obtained based

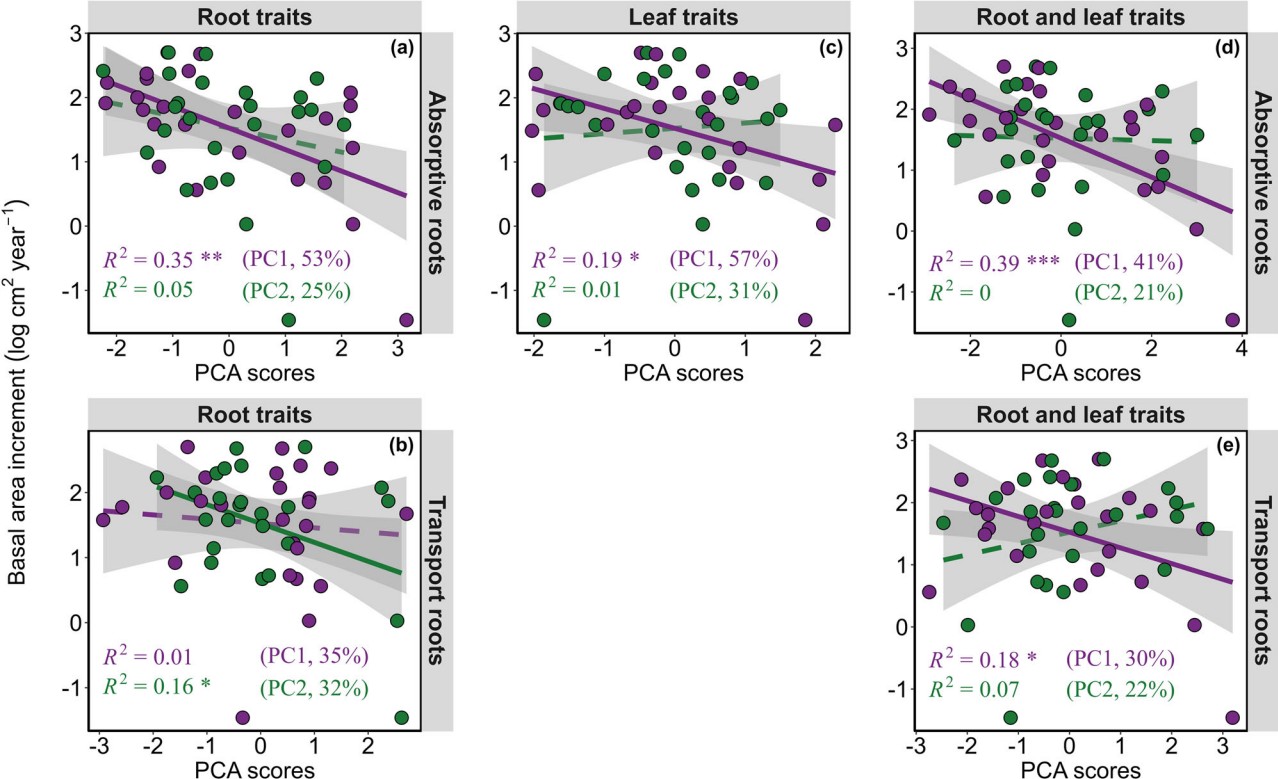

**Fig. 4 | Bivariate plots showing the relationships between average basal area increment and the first and second axes of the principal component analyses.** Relationships between average basal area increment and the first and second axes of the principal component analyses (PCA) of **a** absorptive root traits, **b** transport root traits, **c** leaf traits, **d** absorptive roots and leaf traits, and **e** transport roots and leaf traits. Shown are the $R^2$ and $P$-values of the linear regressions. Significant relationships between basal area increment and PC axes are denoted by * ($P < 0.05$), ** ($P < 0.01$) and *** ($P < 0.001$). Points are colored to distinguish PC1 (purple) from PC2 (green). Solid lines indicate significant relationships at $P < 0.05$ and dashed lines are non-significant relationships at $P > 0.05$. The shadow around the slope corresponds to the 95% confidence intervals.

on single-trait bivariate relationships, where RD, MCR, and cortex-to-stele ratio were significantly and negatively correlated with tree growth, while RTD was positively correlated with tree growth (Fig. S3). In addition, trees in the region were experiencing a drought from 2018 to 2020[53], a higher SRL and smaller RD are associated with higher hydraulic conductivity, which reflects drought tolerance capacity[20,51,54]. So thinner roots, which can potentially access smaller soil pores, are potentially more beneficial, particularly during dry years[51], thereby enabling greater nutrient and water uptake with low resource investment.

Leaf traits were significantly related to tree growth, both alone and when combined with absorptive roots, in support of our third hypothesis (H3), showing that tree growth is best explained by a combination of leaf and root traits. This can be attributed to the fact that above-and below-ground absorptive organs (leaf and absorptive roots) are directly responsible for resource acquisition, showing their significant contributions to tree growth rate. Indeed, tree growth is highly dependent on the concurrent acquisition of above-and below-ground resources, and this can be optimized by the coordination of absorptive roots and leaves. Particularly, leaves play crucial roles in plants by converting sunlight energy, carbon dioxide and water into organic carbon through photosynthesis[5,6,55], thereby influencing growth. Our results corroborate previous studies showing the importance of leaf traits and their contribution to forest functioning[4,56]. Our results showed that thinner absorptive roots that are less colonized with arbuscular mycorrhiza were related to high tree growth, but this was the case only for species with higher LDMC and LMA. More specifically, species with a slow-growing strategy (higher LDMC and LMA) tend to have higher growth, indicating a decoupled root and leaf trait strategy explaining tree growth. The opposite pattern has been reported, where species with a high RD and a higher LMA enhanced tree growth[40]. The higher growth observed in species

with a slow-growing strategy might be due to the fact that slow-growing species are less sensitive to drought than fast-growing species[57]. This is consistent with a current study that reported that slow-growing species exhibited a higher growth resistance to drought[58]. Moreover, the decoupling of leaf and root traits suggests the presence of various adaptive phenotypic strategies that enhance tree growth. This might also indicate that the integration of morphological properties might differ between above- and below-ground organs[59]. This could be attributed to the complexity of below-ground resource uptake, where tree species may possess varying combinations of root traits to effectively explore soil pores and uptake resources. Taken together, these findings confirm that traits more directly related to resource uptake above- and below-ground are important indicators of tree growth[40,49].

While the second PC axis of transport roots, indicating species with a higher cortex-to-stele ratio that are more colonized with arbuscular mycorrhiza, is significantly related to tree growth, its predictive power is only half as strong as that of absorptive roots (Fig. 4a, b). We argue that this might be related to similar mycorrhizal colonization rates in both absorptive and transport roots (Figs. 1d, 3b, 4b, S3j, S4d). In addition, incorporation of leaf traits to transport root traits slightly increased the explanatory power of estimating tree growth. Based on the trait loading on the PC axes (Table S1), we argue that this contribution arises from leaf traits rather than transport root traits, as LDMC and LMA are primarily loaded on the first PC axis (Fig. 3e, Table S1). The smaller explanatory power of transport roots in tree growth compared to absorptive roots confirms that transport roots are mainly involved in the transport and storage of resources and also play crucial roles in protecting plants against pathogens and dehydration[60,61] rather than resource acquisitions that are directly related to growth[11]. Another possible explanation could be that most of the root traits

**Table 2 | Comparison of the models obtained from a series of single and/or multiple linear regression analyses between average basal area increment and principal component (PC) axes**

| | Absorptive roots (Fig. 3a) | | | Transport roots (Fig. 3b) | | | Leaf traits (Fig. 3c) | | | Absorptive roots and leaf traits (Fig. 3d) | | | Transport roots and leaf traits (Fig. 3e) | | |
|---|---|---|---|---|---|---|---|---|---|---|---|---|---|---|---|
| PCA | PCA | R² | AIC | PCA | R² | AIC | PCA | R² | AIC | PCA | R² | AIC | PCA | R² | AIC |
| | PC1 (53%) | **0.35**\*\* | 61.13 | PC1 (35%) | 0.01 | 71.68 | PC1 (57%) | **0.19**\* | 66.59 | PC1 (41%) | **0.39**\*\*\* | 59.38 | PC1 (30%) | **0.18**\* | 66.96 |
| | PC2 (26%) | 0.05 | 70.56 | PC2 (32%) | **0.16**\* | 67.62 | PC2 (31%) | 0.01 | 71.71 | PC2 (21%) | 0.001 | 71.88 | PC2 (22%) | 0.07 | 70.16 |
| | PC3 (16%) | 0.01 | 71.56 | PC3 (19%) | 0.12 | 68.67 | PC3 (13%) | 0.05 | 70.73 | PC3 (12%) | 0.01 | 71.72 | PC3 (18%) | 0.05 | 70.61 |
| | PC4 (5%) | 0 | 71.90 | PC4 (8%) | 0 | 71.90 | | | | PC4 (10%) | 0 | 71.89 | PC4 (12%) | 0.06 | 70.40 |
| | PC1:2 (79%) | **0.40**\*\* | 61.03 | PC1:2 (67%) | 0.17 | 69.35 | PC1:2 (87%) | **0.20**\* | 68.36 | PC1:2 (62%) | **0.39**\*\*\* | 61.35 | PC1:2 (52%) | **0.25**\* | 66.82 |
| | PC1:3 (95%) | **0.42**\*\* | 62.45 | PC1:3 (86%) | **0.29**\* | 67.42 | PC1:3 (100%) | **0.24**\* | 68.89 | PC1:3 (74%) | **0.40**\*\*\* | 63.05 | PC1:3 (70%) | **0.30**\* | 67.09 |
| | PC1:4 (100%) | **0.42**\*\* | 64.45 | PC1:4 (94%) | **0.29**\* | 69.42 | | | | PC1:4 (84%) | **0.40**\*\* | 65.03 | PC1:4 (82%) | **0.36**\* | 66.92 |

$R^2$ and AIC (Akaike Information Criterion) for each model are given. Significant relationships between basal area increment and PC axes are denoted by * ($P < 0.05$), ** ($P < 0.01$) and *** ($P < 0.001$), and $P < 0.05$ are given in bold. Selected models based on ≥70% explained variance are highlighted in gray. Explained variance (%) by PC axis(es) is given in parenthesis.

considered, i.e., RD, RTD, SRL and MCR, are more likely related to resource acquisition[7] than to the hydraulic properties of the transport roots.

By functionally separating fine roots into absorptive and transport roots, our results show a strong association between absorptive fine root traits and broadleaved tree growth in a research arboretum. A higher contribution of absorptive root traits to predicting tree growth suggests that variation in absorptive root traits, rather than transport root traits, better explains tree growth variation by presumably providing resources, e.g., nutrients and water, directly influencing overall tree growth. We argue that by considering fine roots (≤2 mm in diameter) as a homogenous pool, the variance of root traits along root orders might be underestimated and might not clearly show root functioning signals. We also acknowledge that further research assessing the role of root and leaf nutrient concentrations as well as considering transport root-related functions may be particularly illuminating.

## Methods

### Study area and experimental design

This study was carried out in the research arboretum ARBOfun located near Leipzig, Germany (51°16′N, 12°30′E; 150 m a.s.l.). The arboretum was established between 2012 and 2014 and harbors 100 tree species belonging to 39 families planted 5.8 m apart. The 2.5 ha of the arboretum is subdivided into five blocks; in each block, one individual of each species was planted. The mean annual precipitation is approximately 534.3 mm, and the mean annual temperature is 9.4 °C[62]. The soil type of the arboretum, which was previously used as a managed arable field, is Luvisol, and it has a pH of 5.7[63]. Out of 100 species, 25 broadleaved species were chosen for this study based on tree diameter and vitality of at least three individuals per species (Table 1, Fig. S1). We also chose 25 species due to the labor-intensive process involved in collecting and analyzing root samples from different root orders.

### Root sampling and measurement

In 2018 and 2019, the roots of three individuals per species were sampled. First, the soil around the target tree was loosened using a digging fork, and then roots were uncovered carefully by hand and with smaller gardening tools. If a root of higher order was found, it was traced towards the main stem of the target tree to confirm its identity. Then intact root branches containing at least the first five root orders, with the most distal root tip as the first root order, were collected. The root samples, including adherent soil, were wrapped in moist paper, sealed in a plastic bag and stored in a cooling box before being transported to the laboratory. After washing root samples, the sample of each individual tree was divided into two portions: one small portion for examining anatomical traits and another for examining morphological traits. Each subsample comprised fine roots spanning the first to fifth root orders. Finer cleaning was conducted using tweezers under the stereo microscope. After cleaning, the different root orders of the fine root samples were identified and then dissected for trait examination, with each root order being analyzed separately. Dissection of root orders was done under a stereo microscope with a scalpel, starting with the root tips as the first root order and categorizing higher root orders towards the stem[8]. From each root sample, 60 root pieces of the first and second root orders, 20 root pieces of the third root order and 10 root pieces of the fourth and fifth root orders were dissected and stored separately in 1.5 ml Eppendorf tubes with water until further processing. The samples of each root order were scanned using a flatbed scanner (Epson Expression 11000XL, UK) at a resolution of 600 dpi, then root pieces were collected, oven-dried at 60 °C for over 48 h and weighed to obtain the root dry weight.

All morphological root traits were quantified by root orders at the individual tree level using root scans, which were analyzed in a batch using the RhizoVision Explorer[64]. Using the provided data in RhizoVison—mainly the average root diameter (RD), the total root length and root volume —alongside the root dry weight data, RTD (root dry weight/root volume) and SRL (total root length/root dry weight) were calculated.

For the measurement of anatomical root traits, root subsamples were cleaned similarly as above, separated by root orders, and placed in scintillation vials containing fixing solution Roti®-Histofix 4% formaldehyde. The

samples were left at room temperature for two hours and then refrigerated overnight. The next day, root samples were dehydrated with a series of ethanol with steps of 10%, 30%, 50% and 70%, in which the root samples rested for one hour each to gradually remove the water remained in the root tissue[65]. Samples were kept in the refrigerator in another 70% ethanol solution until further processing. We used an automated tissue processing system (Donatello, Diapath) with (i) 45 min each at 38 °C: twice 80% ethanol and twice 96% ethanol, (ii) 60 min each at 38 °C and at 40 °C xylol and (iii) 80 min each at 62 °C three times paraffin, followed by manual embedding of root fragments using a paraffin embedding center (TES 99, Medite). Embedded samples were cross-cut to 1–3 μm with a sledge microtome (DDMP, Medim), put on a slide, processed twice for 10 min in xylol, followed by each in 5 min 96%, 80% and 70% ethanol, and finally distilled water before staining for 2 min in 0.01% toluidine blue solution (Aldrich). Slides were permanently fixed with a Tissue Tek system (Sakura). Then, the images of cross-sections per root order were recorded with a microscope (Axiostar plus, Zeiss, Germany) and a microscope camera accompanied with the program AxioVision (Zeiss, Oberkochen, Germany). We ensured that the entire cross-section as well as a representative section of higher resolution was depicted in the cross-section image. Analysis of the images for measuring root area, stele area, cortex area and cortex area to stele area ratio (C:S ratio) was done with ImageJ[66]. The anatomical differences across root orders for six different tree species can be seen in Fig. S6.

The rate of arbuscular mycorrhizal colonization (MCR) was investigated using the magnified intersection method[46]. Root pieces were bleached in 10% potassium hydroxide for 18 h. Next, roots were rinsed using deionized water and stained in a 10% ink-vinegar solution [67] for 15 min at 90 °C in a water bath. Stained root samples were stored in lactoglycerol until processing. MCR of root pieces was quantified by examining hyphae, arbuscules, hyphal coils, vesicles, and AMF according to the magnified intersection method [46] with a microscope slide at a magnification of 200x.

According to two distinct groups of fine roots based on their functional roles[11], we used the average of the first three root-order traits to represent absorptive roots and the average of the fourth to fifth root-order traits to represent transport roots for further analyses. In order to verify this classification, we ran additional analyses separately across root orders. The results of these additional analyses are presented in supplementary information (Fig. S5), where our main findings and conclusion remain the same, verifying the classification of absorptive and transport roots based on root orders.

## Leaf sampling and measurement
Eleven fully expanded and intact sun-exposed leaves were randomly selected and collected from each individual tree species between 2018 and 2022, following the standard protocol[68]. Of the eleven leaves, five were scanned at 600 dpi with a flatbed Expression 11000XL, and the images were analyzed using WinFolia (Regent Instrument, Canada) to get the fresh leaf area. After scanning, the leaf materials were dried, and weighed to get their fresh weights. Then the samples were oven-dried at 60 °C for five days and weighed. The leaf mass per area (LMA) was computed by dividing the dry mass of the five leaves (including both lamina and petiole) by their total fresh area. The LDMC was determined by dividing the mean leaf dry weight by the mean leaf fresh lamina weight. We measured force to punch using a motorized vertical test stand along with a Sauter FH50 with a dynamometer combined with a flat-sided needle on three positions of three leaves per species. Additionally, three leaves per species were manually crosscut using a blade to obtain thin sections in the central area of the leaf. The resulting cross sections were then placed in a drop of water on an object slide and examined under a microscope. Then, the mean leaf thickness was determined using the Axiocam (Zeiss, Germany) and the software ZEN 2 core. We then calculated the leaf toughness (LT) for each leaf by dividing force to punch of the leaf by the leaf thickness and then computed the individual mean LT[69].

## Quantification of tree growth
The diameter at breast height (DBH) of each individual tree species was measured using a calliper each year, except 2020. We then calculated basal area increment as a proxy for tree growth using the sum of DBH data for individual tree species. As such, we calculated the average absolute basal area increment by dividing the 2022 basal area data of each individual tree by its age since planting. Hence, the average basal area increment was calculated according to the following equation:

$$\text{Average basal area increment} = ln\left(\frac{\sum_{j=1}^{n}\left(\frac{\pi}{4} * DBH_{j(2022)}^2\right)}{\text{tree age since planting}}\right)$$

where DBH is the diameter at breath height measured at the 1.3-m height of an individual, $j$ is an index for the $n$ stems of the individual, and 2022 is the year when DBH of the individual tree was measured, which overlap the years (2018–2022) during which the trait measurements were done.

## Statistical analyses
To assess the variation and coordination of the absorptive and transport root traits, we performed principal component analyses (PCAs) using stepwise inclusion of root and leaf traits at the species average level. To do so, the first set of PCA were performed on morphological and anatomical root traits for absorptive and transport roots separately. The second PCA was performed on leaf traits (LDMC, LMA, and LT). Finally, we performed a third set of PCAs on integrated root traits as well as leaf traits. The PCAs were performed on scaled trait data and without axis rotation. To aid interpretation, we inverted the PCA axis of the transport root traits by multiplying by minus one whenever required. After each PCA, we investigated for each of the raw trait variables whether it was most related to either the first or second PCA axis. For example, if a given trait had a loading of +0.41 on PC1, and −0.45 on PC2, we concluded it was most related to PC2, given the higher absolute loading value. Specifically, we performed linear regression to quantify the relationships between average basal area increment (as a dependent variable) and the first and second PC axes scores (as the explanatory variables) of each PCA coordination. We conducted regression analyses, incorporating both single and multiple PC axes as explanatory predictors for tree growth, until the cumulative explained variance for each PCA group reached 70%. As the first two PC axes captured most of the variance, we only showed the bivariate regression graphs between those PC axes and average basal area increment. Additionally, we used a paired two-sample t-test to compare root traits between absorptive and transport roots. To complement the results of PCAs on traits, we subsequently explored the pairwise correlations by performing Pearson's correlations between absorptive or transport root traits and leaf traits using the ggraph function of the 'ggraph' package[70]. To assess each single root and leaf trait as an explanatory predictor for tree growth, we further performed bivariate linear regression separately across absorptive or transport root and leaf traits. To meet the linear regression assumptions, the average basal area increment was log-transformed before the regression analysis.

## Statistics and reproducibility
We studied 25 tree species with three replicates each in the research arboretum ARBOfun located near Leipzig, Germany. The sampled root traits were classified into two distinct groups of fine roots based on their functional roles: the average of the first three root-order traits as absorptive roots and the average of the fourth to fifth root-order traits as transport roots. We then performed the PCAs using the prcomp function of the 'stats' package using the mean of absorptive and transport roots as well as leaf traits across 25 species. Using the lm function in the "stats" package, we then performed linear regression to quantify the relationships between average basal area increment and the first and second PC axes scores of each PCA coordination. All analyses were done using the R v.4.3.2 platform[71].

## Reporting summary
Further information on research design is available in the Nature Portfolio Reporting Summary linked to this article.

## Data availability
The data [72] that support the findings of this study are openly available in the iDiv Data Repository.

## Code availability
The source code [72] for reproducing the results is available in the iDiv Data Repository.

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

## Acknowledgements

We thank Roman Patzak, Imke Pelloth, Lea von Sivers, Tom Künne and Julia Leonore van Braak for their help with the field and lab measurements. We also thank Maritta Wipplinger from the institute of veterinary pathology at Leipzig University for her help in root cross-section preparation. We are especially grateful to Florian Schnabel, Lena Kretz and David Schellenberger Costa for helping with discussing ideas. We thank the iDiv Data & Code Unit for assistance with curation and archiving of the dataset. A.S. is supported by the Saxon State Ministry for Science, Culture and Tourism (SMWK) – [3-7304/35/6-2021/48880].

## Author contributions

A.W., C.W., and A.S. conceived the ideas and developed the concept of the study. A.S., F.v.d.P., H.C., S.D., S.E., A.K., Y.M., J.S., and A.W. contributed to data collection. A.S. analyzed the data and led the writing of the manuscript. F.v.d.P., H.C., R.R., J.H., C.W., and A.W. contributed to the writing in several manuscript interactions. All authors contributed critically to the drafts and gave final approval for publication.

## Funding

## Competing interests

The authors declare no competing interests.
