## [Transparent Peer Review file · Communications Biology]

Tree growth is better explained by absorptive fine root traits than by transport fine root traits

Corresponding Author: Dr Anvar Sanaei

Version 0:

Reviewer comments:

Reviewer #1

(Remarks to the Author)

This study aims to answer the question whether functional traits of absorptive (orders 1-3) or transport (orders 4-5) fine roots of temperate broadleaf trees explain more variation in basal area increment. Based on several principal component analyses (PCA) and subsequent linear models with the resulting principal components as predictors for growth, the authors conclude that the traits of absorptive fine roots are more important for growth.

The approach is intriguing and new, and generally suitable to adequately answer the research questions. A strength of the experimental design is that all trees are roughly the same age and grow under the same site conditions, excluding possible confounding factors. The paper is nicely written and a good read. I still have a few bigger and smaller issues that have to be addressed before the paper can be considered suitable for publication.

When interpreting the PCAs, it is very subjective and often not comprehensible why the authors attribute certain traits to certain axes. For example, they state in line 180 that for the absorptive roots-leaves PCA, LDMC is associated with PC1, while the respective figure shows that LDMC is about equally correlated with PC1 and PC2. It would be crucial to use an objective criterion, like e.g. the correlations between PCs and traits.

I think the paper would benefit if the data analysis was done in a slightly different way. As most of the PCA axes did not turn out to be very clear proxies of specific sets of traits anyway and are therefore of little informative value by themselves, I suggest to fit linear models where PC1 and PC2 (and ideally even more PCs, until their cumulative explained variance reaches satisfactory, comparable levels) of each PCA are included simultaneously instead of two single linear models. This way, the R2 would be interpretable as the total explained variance by e.g. absorptive root traits, and R2 values could be compared directly. Comparing how well PC1 of the absorptive root PCA and PC1 of the transport root PCA explain growth separately does not make any sense because these axes don't represent the same functional spectrums.

The figures do not fulfill aesthetic standards and could be more informative if a few changes, which I suggest in detail below, were made. Most importantly, Fig. 4, showing the central result of the paper, has to be more interpretable.

Specific comments:

Intro:

69: Why are "fine roots" in quotation marks?

70: RES conventionally stands for root economics spectrum, not space

74: High root nitrogen content

82: The fact that different root orders can be structurally and anatomically different is already pointed out in the previous sentence

84: There is no consensus that root order 3 is a universal threshold for absorptive/transportive roots, which orders predominantly perform which functions can also vary between species

85: Remove "soil-based" (unnecessary)

85: "...biotic interactions with soil microbes and mycorrhiza"

89: Instead of "tied to the location within the branching root system": Depend on the root order

98: CS ratio is not directly an indication of these functions, I think it would be more precise to add "...an indication of the capacity for resource absorption..."

100: This sentence sounds like this is new evidence, but secondary growth and the associated anatomical changes in roots have been known for a long time and are not debated. Also, when there is no cortex, colonization by arbuscular mycorrhiza is not only reduced but absolutely not possible at all

106-108: This sentence is unnecessarily circuitous while at the same time not saying much. What exactly is "functional coordination" supposed to mean?

115: If the links are weak in literature, why is it so important then? Also, two times "link" in one sentence

141: The applied root order classes are almost certainly not functionally discrete, the degree of absorptive and transportive functioning is a continuous gradient over root orders

152: But why wouldn't transport be important as well? In Central European forests, tree growth is often water-limited, so one

would think that hydraulic properties of transport roots matter a lot as well. Especially since these trees lived during 2018-2020. Like this, it sounds like a post-hoc hypothesis that was postulated after seeing the results

Methods:

333: Instead of "is designed for": "harbors"

339: Out of the 100 species in the arboretum, why did you choose these 25 species? Were there any objective criteria? Also, the species number is not mentioned here, only in the abstract

435: Did you also log-transform the traits? This is often necessary to fulfill the assumptions of PCA

Results:

162: Instead of "showed a much stronger negative correlation": Were much more negative

173: Collaboration gradient: That's already an interpretation

Discussion:

221: I don't think "whereby" is a fitting word in this context, if the following part of the sentence refers to the collaboration gradient, "where" or "in which" might be better

Figures:

Fig. 1: A similar type of figure could give an additional level of information and be also more suitable to visualize the applied statistical test, the paired t-test: Instead of bar plots, you could plot each species' absorptive and transport root traits as a pair of points, connected with a line. This way, one could also see to what degree the strength and direction of the differences varies between species.

Fig. 3: The panels don't seem to be arranged in an even grid, the spaces between the panels appear off. The box with the legend in text form (suboptimal, a graphical legend would be better) and abbreviations has a weird frame with 3 out of 4 lines. Generally, the abbreviations would better off in the caption. Sometimes, a dash is used, sometimes an equals sign. It would be super interesting to know where each of the species is located in the trait space. For that, you could add small text labels at least for the more outlying species that are not so much clustered in the center.

Fig. 4: It's really hard to interpret this figure without having to Fig. 3 right next to it and looking back and forth between both figures all the time. It would help a big deal if you could add interpretation aids to each of the principal components that are on the x axes in this figure, like for example here: <https://doi.org/10.1111/1365-2435.14554>. Again, the use of dashes and equals signs is not consistent. The font size of the text in the lower middle panel is too small, and again, its information should actually be in the caption. Again, it would be interesting to know which were the fast- and slow-growing species.

Supplement:

Table S1: Two entries are *Acer platanoides*

Reviewer #2

(Remarks to the Author)

The paper presents interesting results and is quite fluently written. It tries to establish which fine root trait coordination originating from PCA analysis better explains tree growth performance. The Authors succeeded in this task, but several important points in the M&M, Results and Discussion sections must be addressed to support the proposed explanations. In some points in the text (not the figures), absorptive and transport roots or the sign of the variable loadings are inverted. The text needs a minor but accurate English revision. Despite the importance of the Cortex-to-Stele ratio as an explanatory variable in the PCA analysis performed, some images of cross-sections of absorptive and transport roots would improve the quality of the paper. I understand the species investigated are 25, but some cross-sections (see comment below) of the more distant species emerging along the gradient could be very explanatory.

The paper needs to be improved before it is eligible for publication. The Results are very interesting, so I strongly encourage the authors to resubmit an improved version. I hope that the above and the following comments may help improve the manuscript.

The importance is the proper use of the terms throughout the text, in particular, "trait coordination"

L40. "...that, for the selected traits,..."

L75. Is low RTD for slow growth? It should be the opposite.

L145-. From this line on, always use the words "trait coordination" as written in L141 to improve clarity and consistency throughout the text.

L157. Results section. Some images of cross-sections of absorptive and transport roots would improve the quality of the paper. I understand you have 25 species, but 6 or 8 cross-sections (3-4 absorptive plus 3-4 transport) of the more distant species emerging along the gradient could be enough and explanatory.

L168-174. SRL was negatively associated on PCA1 both for absorptive and transport roots.

L183-186. Comments are incorrect as all positive and negative associations are inverted; for example, PC1 is positively related to MCR... Why did it happen? Is it because you multiply by -1 whenever required (L435-437)?

L199-205. PC1 axes of absorptive root-leaf and of transport root-leaf are very similar. Why did you comment in the opposite way, i.e., higher MCR and lower growth for absorptive but lower MCR and higher growth for transport roots?

L214. Improve English: "... is better explained by absorptive than by transport root traits and was higher in species with thinner root diameters that were..."

L238. Within the first-order root segments, you have both pioneer and fibrous roots, the former generally with a larger diameter and poorly mycorrhized. Did you find these differences within the absorptive roots? Could these two categories explain part of the observed variance?

L287-289. Please rewrite. Move the sentence 290-291 to L287 after 2020.

L303. A lower SLA corresponds to a higher LMA. Since you were referring to LMA in the previous lines, using the same variable should improve clarity.

L308. "... as that of absorptive roots". Figure 4a refers to absorptive roots.

L309 and 313. Remove “believe”, this is science. I suggest “argue”.

L309-311. Please rewrite this sentence to make it more consecutive with the previous one. For example: “Therefore, we argue that... transport roots, although to a lesser extent for the latter.”

L352-353. Dissection criteria are not exhaustive. Is your hierarchical approach equal-similar to that of Pregitzer et al. 2002 (8)? In Figure 2 of the cited paper, there is a very clear sketch of a small portion of an intact segment of the branching fine root system. I suggest adding a similar picture if your approach was different or referring to this paper if your approach was the same.

L361. Insert the acronym of average root diameter when first mentioned (RD).

L382-383. Why did you not analyse these anatomical traits, root, stele and cortex area separately? Please, provide an justification.

L399-415. Two leaves are missing. You collected 13 and then used 5 + 3 + 3.

L408. What is the “force to punch”? What does the term “punch” mean-specify? Is it a...

L413. Again, insert the acronym of average root diameter when first mentioned (LT).

L417. It may be my fault, but I do not understand why you measured DBH in 2023 and successively, you wrote “2022 basal area data” in L420.

Figure 3. To improve the readability, scale to the same range values all the PCA axes for all panels, precisely -4 to +4 (may be -3.5 to +3.5 is already suitable).

Figure 4. ... the first and second axes of the PCA scores...

As for Figure 3, scale the PCA axes to the same range of values.

Reviewer #3

(Remarks to the Author)

This manuscript, titled “Tree growth is better explained by absorptive fine root traits than by transport fine root traits”, was conducted on 25 European broad-leaved tree species, and examined a suit of morphological and anatomical traits of fine roots (both absorptive and transport roots) as well as leaf traits. This study mainly investigated the coordination of root traits within absorptive and transport roots, and how they help to explain tree growth. Overall, the manuscript is well written and clearly presented, and the findings are also relatively important and timely. However, there are still some points that have not been fully clarified.

1. Your research includes the relationship between leaf traits and tree growth, and your topic is the comparison of absorptive roots and transport roots in explaining tree growth. So, what role do leaf traits play and why does it appear? This needs to be emphasized and clarified in the introduction and discussion.

2. Lines 235-238: Which method did you use? Please clarify (and also include it in the methods section). Additionally, what is your intention in comparing the two methods here? Compared to the morphometric root order classification method, the topological root order classification method defines relatively more transport roots. However, this seems to deviate too much from what you are trying to explain.

3. The mycorrhizal colonization rate of transport roots is quite close to that of absorptive roots, which seems counterintuitive. Even if it is species-specific, theoretically, the colonization rate should gradually decrease with the increase in root order, considering the transition from absorption to transport function. Since you have colonization rate data for each root order, you could add such data analysis and presentation (i.e., the variation pattern of colonization rate with root order and the differences).

4. In addition, considering the species specificity, if the colonization rate of the 4/5-order roots in your study is similar to that of the absorptive roots, does this mean that in the tree population you studied, the higher-level root orders should be the transport roots? In other words, without looking at the anatomy, the method of directly dividing the absorptive vs transport roots by root order has great limitations, which need to be pointed out in the paper.

5. Trees with high growth rates are theoretically expected to correspond to traits related to a fast-slow strategy dimension. However, in this case, the traits are more closely related to mycorrhizal traits, especially mycorrhizal colonization rates. Additionally, both composite and individual leaf traits are positively correlated with average basal area increment (which reflects tree growth). These two results seem somewhat counterintuitive, so you may need to explore these issues further in your discussion.

6. The clarification on whether coordination exists between absorptive roots and transport roots is still insufficient. This is evident in two aspects: first, the presentation of data, and second, the discussion in your manuscript. A possible approach to address this could be to supplement your analysis by including both absorptive and transport root traits in a PCA to observe the trait distribution patterns.

7. Lines 300-302: The slow strategy, based on leaf traits, corresponds to higher tree growth rates, suggesting a decoupling between roots and leaves in explaining tree growth. This raises several unclear issues: the alignment and directionality of the strategy dimension represented by traits in explaining tree growth, and the inconsistency among different organs. These issues need to be better clarified and discussed in more detail.

8. Lines 155-156: Would it be more appropriate to focus on absorptive roots in this hypothesis, considering that absorptive

roots, like leaves, are organs responsible for resource acquisition?

9. Line 195 (Figure 2d) and Line 314 (Figure 1f). There are many similar figure citation errors in the full text, which will reduce the readability of the paper. Please check it throughout the text.

Version 1:

Reviewer comments:

Reviewer #1

(Remarks to the Author)

I am glad to say that all my concerns have been adequately addressed. The data analysis and the figures have been revised according to my suggestions. From my perspective, the manuscript can now be accepted. Congratulations to the authors for the great work!

Reviewer #2

(Remarks to the Author)

The paper is actually improved and all my raised comments have been addressed. In my opinion, it is now ready for publication.

Colours are not explained in Figure 3

Reviewer #3

(Remarks to the Author)

The authors have already addressed most of the issues I raised, but there are still a few issues I would like to provide feedback to the authors.

1. Regarding the fourth question, specifically lines 270-285, I believe the discussion here could be more concise. Additionally, it might be more appropriate to adopt a tone that frames it as "future research needs to...".

2. Regarding the fifth question, your answer "the bivariate regressions showed that tree growth was negatively correlated with root diameter (RD), cortex to stele ratio (C:S) and mycorrhizal colonization rates (MCR) (Figure S3a,d,e). We found similar results with the same directionality in the PCA scores-growth relationships, where tree species with thicker roots and lower MCR grew better (Figures 3a,4a)."

Please carefully verify the accuracy and consistency of the descriptions and figures here, and everywhere else in the maintext except in this reply.

3. Regarding the sixth question, specifically lines 241-244, please carefully check the results in Fig. 3 and Fig. S2, and clarify which of your results are similar to those of previous studies. Does your Fig. 3b support the claim that species with larger root diameters were highly related to mycorrhizal association?

Reviewers' comments:

Reviewer #1:

This study aims to answer the question whether functional traits of absorptive (orders 1-3) or transport (orders 4-5) fine roots of temperate broadleaf trees explain more variation in basal area increment. Based on several principal component analyses (PCA) and subsequent linear models with the resulting principal components as predictors for growth, the authors conclude that the traits of absorptive fine roots are more important for growth.

The approach is intriguing and new, and generally suitable to adequately answer the research questions. A strength of the experimental design is that all trees are roughly the same age and grow under the same site conditions, excluding possible confounding factors. The paper is nicely written and a good read. I still have a few bigger and smaller issues that have to be addressed before the paper can be considered suitable for publication.

Response: Thank you for your positive feedback on our manuscript and the many valuable comments you provided. We tried to address all of these comments with great care, and provide point by point responses below.

When interpreting the PCAs, it is very subjective and often not comprehensible why the authors attribute certain traits to certain axes. For example, they state in line 180 that for the absorptive roots-leaves PCA, LDMC is associated with PC1, while the respective figure shows that LDMC is about equally correlated with PC1 and PC2. It would be crucial to use an objective criterion, like e.g. the correlations between PCs and traits.

Response: We agree that PCA interpretation is somehow subjective. We assigned certain traits to the highest PCA score of that trait (Table S1, see also lines 491-494). For example, based on the PCA scores presented in Table S1, the absolute score of LDMC on the first PC axis (-0.42) is greater than the absolute score of LDMC on the second PC axis (-0.31). In such cases, we assigned the trait to the greater absolute scores of the PC. Bivariate correlation between LDMC with PC scores showed that there was only significance between PC1 and LDMC ($R^2=0.57^{**}$) but not PC2 with ($R^2=0.16$).

I think the paper would benefit if the data analysis was done in a slightly different way. As most of the PCA axes did not turn out to be very clear proxies of specific sets of traits anyway and are therefore of little informative value by themselves, I suggest to fit linear models where PC1 and PC2 (and ideally even more PCs, until their cumulative explained variance reaches satisfactory, comparable levels) of each PCA are included simultaneously instead of two single linear models. This way, the R^2 would be interpretable as the total explained variance by e.g. absorptive root traits, and R^2 values could be compared directly. Comparing how well PC1 of the absorptive root PCA and PC1 of the transport root PCA explain growth separately does not make any sense because these axes don't represent the same functional spectrums.

Response: Thank you for the great suggestion; you make a very good point. We agree with the reviewer that simultaneously fitting both PCA axes is more appropriate and provides more information about trait-growth relationships. We have now combined both PC axes information in a single panel (Figure 4). As suggested, we also ran additional analysis, including more PC axes, until their cumulative explained variance reaches 70%, and accordingly updated the results. The results of this updated analysis is represented in Table 2.

The figures do not fulfill aesthetic standards and could be more informative if a few changes, which I suggest in detail below, were made. Most importantly, Fig. 4, showing the central result of the paper, has to be more interpretable.

Response: Thank you for the suggestion; we have now changed the Figures to make them clearer.

Specific comments:

Intro:

69: Why are “fine roots” in quotation marks?

Response: We have now deleted the quotation marks.

70: RES conventionally stands for root economics spectrum, not space

Response: We are here referring to Bergman et al. 2020, where they used RES to mean root economic space.

74: High root nitrogen content

Response: Revised as suggested.

82: The fact that different root orders can be structurally and anatomically different is already pointed out in the previous sentence

Response: Revised as suggested.

84: There is no consensus that root order 3 is a universal threshold for absorptive/transportive roots, which orders predominantly perform which functions can also vary between species

Response: We agree that the functional classification of roots (absorptive and transport roots) is highly species-specific. Considering this comment, we have now added sentences in this context in the Introduction section (Lines 91-92).

85: Remove “soil-based” (unnecessary)

Response: Deleted.

85: “...biotic interactions with soil microbes and mycorrhiza”

Response: Changed as suggested (Line 88).

89: Instead of “tied to the location within the branching root system”: Depend on the root order

Response: Changed as suggested (Line 93).

98: CS ratio is not directly an indication of these functions, I think it would be more precise to add “...an indication of the capacity for resource absorption...”

Response: Changed as suggested (Line 101).

100: This sentence sounds like this is new evidence, but secondary growth and the associated anatomical changes in roots have been known for a long time and are not debated. Also, when there is no cortex, colonization by arbuscular mycorrhiza is not only reduced but absolutely not possible at all

Response: We have now revised this sentence (Lines 103-107).

106-108: This sentence is unnecessarily circuitous while at the same time not saying much. What exactly is “functional coordination” supposed to mean?

Response: Deleted.

115: If the links are weak in literature, why is it so important then? Also, two times “link” in one sentence

Response: We have discussed (Lines 124-129) the reasons for the weak linkage between traits and ecosystem functioning. One reason is the focus on a single organ trait or solely on morphological traits. In this study, we examined both leaf and root traits, and for roots, we considered not only morphological traits but also anatomical traits. We replaced the second ‘link’ with ‘association’ (Line 121).

141: The applied root order classes are almost certainly not functionally discrete, the degree of absorptive and transportive functioning is a continuous gradient over root orders

Response: As mentioned earlier, we agree that the functional classification of roots (absorptive and transport roots) is species-specific and that these changes might occur gradually. However, the order-based PCA and regression analyses results (Figure S5) are almost consistent with absorptive and transport roots classification results (Figures 3,4).

152: But why wouldn't transport be important as well? In Central European forests, tree growth is often water-limited, so one would think that hydraulic properties of transport roots matter a lot as well. Especially since these trees lived during 2018-2020. Like this, it sounds like a post-hoc hypothesis that was postulated after seeing the results

Response: We appreciate the reviewer's comment. Considering your comment, we have now revised our second hypothesis (Lines 156-158).

Methods:

333: Instead of “is designed for”: “harbors”

Response: Changed as suggested.

339: Out of the 100 species in the arboretum, why did you choose these 25 species? Were there any objective criteria? Also, the species number is not mentioned here, only in the abstract

Response: We have now added a justification for the species selection and the species number mentioned in the Methods section (Lines 386-389). We have now moved the species list information table (Table 1) to the main file. A figure of a phylogenetic tree of the species also added to the supplementary file (Figure S1).

435: Did you also log-transform the traits? This is often necessary to fulfill the assumptions of PCA

Response: Upon checking the relationships between traits, we found that data transformation was unnecessary. However, to ensure data normality, we tested different transformation methods (such as log and square root transformations) but found they did not improve data normality. So, we only Z-transformed the trait data before the PCA analysis. We mentioned that the PCAs were performed using the *prcomp* function of the ‘stats’ package on **scaled** trait data.

Results:

162: Instead of “showed a much stronger negative correlation”: Were much more negative

Response: Corrected as suggested (Lines 170-171).

173: Collaboration gradient: That’s already an interpretation

Response: deleted from the results section.

Discussion:

221: I don’t think “whereby” is a fitting word in this context, if the following part of the sentence refers to the collaboration gradient, “where” or “in which” might be better

Response: Changed as suggested.

Figures:

Fig. 1: A similar type of figure could give an additional level of information and be also more suitable to visualize the applied statistical test, the paired t-test: Instead of bar plots, you could plot each species' absorptive and transport root traits as a pair of points, connected with a line. This way, one could also see to what degree the strength and direction of the differences varies between species.

Response: Thank you for this useful suggestion. We have now revised Figure 1 and included species-specific information in this Figure, where the direction of species from absorptive to transport roots is shown.

Fig. 3: The panels don't seem to be arranged in an even grid, the spaces between the panels appear off. The box with the legend in text form (suboptimal, a graphical legend would be better) and abbreviations has a weird frame with 3 out of 4 lines. Generally, the abbreviations would better off in the caption. Sometimes, a dash is used, sometimes an equals sign. It would be super interesting to know where each of the species is located in the trait space. For that, you could add small text labels at least for the more outlying species that are not so much clustered in the center.

Response: We agree; thanks for pointing out this issue to us. We have now completely revised Figure 3 and moved the legend text from it to the caption.

Fig. 4: It's really hard to interpret this figure without having to Fig. 3 right next to it and looking back and forth between both figures all the time. It would help a big deal if you could add interpretation aids to each of the principal components that are on the x axes in this figure, like for example here: <https://doi.org/10.1111/1365-2435.14554>. Again, the use of dashes and equals signs is not consistent. The font size of the text in the lower middle panel is too small, and again, its information should actually be in the caption. Again, it would be interesting to know which were the fast- and slow-growing species.

Response: Thank you for this useful suggestion. We agree with the reviewer; however, in our case, it's not that easy to add interpretation aids. First of all, as we have now combined the two PC axes in a single panel, adding the interpretation aids to the two PC axes shown in a single panel might lead to confusion in interpreting the results (Figure 4). Second, for leaves, we only have conservative leaf traits (leaf dry matter content, leaf mass per area, and leaf toughness), not acquisitive leaf traits (leaf nitrogen and specific leaf area). For roots, we do not have root nitrogen to assign any root axis to the root conservation gradient (root nitrogen-root tissue density trade-off), and also the root collaboration gradient (root diameter-specific root length trade-off) is collapsed in some cases due to the missing the fourth trait (Root nitrogen). So, we really cannot clearly assign the axes to any trade-off gradients.

As mentioned earlier, we have now revised Figures 3 and 4, showing species information within PCA.

Supplement:

Table S1: Two entries are *Acer platanoides*

Response: Corrected.

Reviewer #2:

The paper presents interesting results and is quite fluently written. It tries to establish which fine root trait coordination originating from PCA analysis better explains tree growth performance. The Authors succeeded in this task, but several important points in the M&M, Results and Discussion sections must be addressed to support the proposed explanations.

Response: Thank you for your positive feedback on our manuscript and the many valuable comments you provided. We tried to address all of these comments with great care, and provide point by point responses below.

In some points in the text (not the figures), absorptive and transport roots or the sign of the variable loadings are inverted. The text needs a minor but accurate English revision.

Response: We have now double-checked the Figure citation and the direction of the trait loadings.

Despite the importance of the Cortex-to-Stele ratio as an explanatory variable in the PCA analysis performed, some images of cross-sections of absorptive and transport roots would improve the quality of the paper. I understand the species investigated are 25, but some cross-sections (see comment below) of the more distant species emerging along the gradient could be very explanatory.

Response: We agree; thanks for pointing out this issue to us. We have now provided a Figure in the supplementary file having cross-sectional images for six tree species (Figure S6).

The paper needs to be improved before it is eligible for publication. The Results are very interesting, so I strongly encourage the authors to resubmit an improved version. I hope that the above and the following comments may help improve the manuscript.

Response: We very much appreciate that the reviewer found our paper of interest.

The importance is the proper use of the terms throughout the text, in particular, “trait coordination”

Response: Good point. We have now kept the consistency of using trait coordination.

L40. “...that, for the selected traits,...”

Response: revised as suggested.

L75. Is low RTD for slow growth? It should be the opposite.

Response: We apologize – this statement was indeed incorrect, and we have now corrected it (Line 79).

L145-. From this line on, always use the words “trait coordination” as written in L141 to improve clarity and consistency throughout the text.

Response: Good point. We have now kept the consistency of using trait coordination.

L157. Results section. Some images of cross-sections of absorptive and transport roots would improve the quality of the paper. I understand you have 25 species, but 6 or 8 cross-sections (3-4 absorptive plus 3-4 transport) of the more distant species emerging along the gradient could be enough and explanatory.

Response: We agree; thanks for pointing out this issue to us. As mentioned earlier, we have now provided a Figure in the supplementary file (Figure S6) having cross-sectional images for six tree species.

L168-174. SRL was negatively associated on PCA1 both for absorptive and transport roots.

Response: Based on the PCA scores of the traits represented in Table S1, for absorptive roots, SRL is more strongly positively loaded on the second axis (0.55) and, of course, with a weaker negative loading on the first axes as well (-0.46), while for transport roots, SRL is negatively loaded on the first PC axis (-0.69). Bivariate regression between PC axis and SRL showed that SRL for absorptive roots significantly ($P < 0.001$) was correlated with both PC axes, while for transport roots, SRL was only significantly correlated with PC1. We have now added an explanation on how we assigned a certain trait to a specific axis (Lines 491-494).

L183-186. Comments are incorrect as all positive and negative associations are inverted; for example, PC1 is positively related to MCR... Why did it happen? Is it because you multiply by -1 whenever required (L435-437)?

Response: That's true. To make the PCA more comparable, both the first and second PCA axes of the transport roots-leaf traits (Figure 3e and Table S1) are inverted by multiplying -1. That's why we reported them in the opposite way; however, the arrows within Figure 3e pointing in the same direction. To make them clearer, we have now mentioned, within the main text (Line 193) as well as in the Figure 3 caption, that the two PC axes in Figure 3e are inverted. Additionally, to make it more clearer and match Figure 3 and Table S1 results, we have now inverted the two PC axes loading scores in Table S1 as well. Following this adjustments, we have now reported the inverted results (Lines 191-193).

L199-205. PC1 axes of absorptive root-leaf and of transport root-leaf are very similar. Why did you comment in the opposite way, i.e., higher MCR and lower growth for absorptive but lower MCR and higher growth for transport roots?

Response: As mentioned above, to make the PCA more comparable, both the first and second PC axes of the transport roots and leaf traits (Figure 3e and Table S1) are inverted. That's why our reported results were not in the same direction. As mentioned above, to make it more clearer we have now reported the direction of the results for both absorptive and transport roots (trait-growth relationships) in the same way (Lines 212-218).

L214. Improve English: "... is better explained by absorptive than by transport root traits and was higher in species with thinner root diameters that were..."

Response: Revised as suggested.

L238. Within the first-order root segments, you have both pioneer and fibrous roots, the former generally with a larger diameter and poorly mycorrhized. Did you find these differences within the absorptive roots? Could these two categories explain part of the observed variance?

Response: We agree, but we did not look into the differences between pioneer and fibrous roots unfortunately as it was not the focus of our study.

L287-289. Please rewrite. Move the sentence 290-291 to L287 after 2020.

Response: Revised as suggested (Lines 321-325).

L303. A lower SLA corresponds to a higher LMA. Since you were referring to LMA in the previous lines, using the same variable should improve clarity.

Response: Revised as suggested (Line 341).

L308. "... as that of absorptive roots". Figure 4a refers to absorptive roots.

Response: We have now double checked Figure citation throughout the text.

L309 and 313. Remove "believe", this is science. I suggest "argue".

Response: Revised as suggested.

L309-311. Please rewrite this sentence to make it more consecutive with the previous one. For example: "Therefore, we argue that... transport roots, although to a lesser extent for the latter."

Response: Revised as suggested (Lines 356-357).

L352-353. Dissection criteria are not exhaustive. Is your hierarchical approach equal-similar to that of Pregitzer et al. 2002 (8)? In Figure 2 of the cited paper, there is a very clear sketch of a small portion of an intact segment of the branching fine root system. I suggest adding a similar picture if your approach was different or referring to this paper if your approach was the same.

Response: We have now cited the Pregitzer paper to the corresponding paragraph as we in principle followed the same procedure.

L361. Insert the acronym of average root diameter when first mentioned (RD).

Response: Revised as suggested.

L382-383. Why did you not analyse these anatomical traits, root, stele and cortex area separately? Please, provide an justification.

Response: Instead of having more traits with the same information, we restricted ourselves to including the cortex-to-stele ratio (C:S), as it combined both cortex and stele information in a single trait.

L399-415. Two leaves are missing. You collected 13 and then used 5 + 3 + 3.

Response: there was a typo mistake, corrected.

L408. What is the “force to punch”? What does the term “punch” mean-specify? Is it a...

Response: Force to punch is the resistance of the tissue to a penetrometer measured in N. The penetrometer used in this study has a flat-ended punch of ~1mm diameter attached to a power stand which records the force needed to punch a hole into the tissue. “Force to punch” is the term used for this measure in standardized trait protocols (Perez-Harguideguy et al. (2013)).

Perez-Harguideguy et al. 2013. New handbook for standardised measurement of plant functional traits worldwide. *Australian Journal of Botany*, 1-70. <https://dx.doi.org/10.1071/BT12225>

L413. Again, insert the acronym of average root diameter when first mentioned (LT).

Response: revised as suggested.

L417. It may be my fault, but I do not understand why you measured DBH in 2023 and successively, you wrote “2022 basal area data” in L420.

Response: We think there is a misunderstanding here that might arise from the unclear explanation of inventory measurement. We now more clearly explained this point in the revised version of the manuscript (Lines 471-472).

Figure 3. To improve the readability, scale to the same range values all the PCA axes for all panels, precisely -4 to +4 (may be -3.5 to +3.5 is already suitable).

Response: All figures revised as suggested.

Figure 4. ... the first and second axes of the PCA scores...

Response: revised as suggested.

As for Figure 3, scale the PCA axes to the same range of values.

Response: Figures 3 revised as suggested.

Reviewer #3:

This manuscript, titled “Tree growth is better explained by absorptive fine root traits than by transport fine root traits”, was conducted on 25 European broad-leaved tree species, and examined a suit of morphological and anatomical traits of fine roots (both absorptive and transport roots) as well as leaf traits. This study mainly investigated the coordination of root traits within absorptive and transport roots, and how they help to explain tree growth. Overall, the manuscript is well written and clearly presented, and the findings are also relatively important and timely. However, there are still some points that have not been fully clarified.

Response: We thank the reviewer for the positive assessment of our study and for the constructive feedback, which has improved the manuscript.

1. Your research includes the relationship between leaf traits and tree growth, and your topic is the comparison of absorptive roots and transport roots in explaining tree growth. So, what role do leaf traits play and why does it appear? This needs to be emphasized and clarified in the introduction and discussion.

Response: This is a great point, thanks for bringing up this point. We have now addressed this point in the Introduction (Lines 69-71) and Discussion (Lines 328-332) sections.

2. Lines 235-238: Which method did you use? Please clarify (and also include it in the methods section). Additionally, what is your intention in comparing the two methods here? Compared to the morphometric root order classification method, the topological root order classification method defines relatively more transport roots. However, this seems to deviate too much from what you are trying to explain.

Response: We apologize – this statement was indeed incorrect, and we have now corrected it and we have now further explained this context.

3. The mycorrhizal colonization rate of transport roots is quite close to that of absorptive roots, which seems counterintuitive. Even if it is species-specific, theoretically, the colonization rate should gradually decrease with the increase in root order, considering the transition from absorption to transport function. Since you have colonization rate data for each root order, you could add such data analysis and presentation (i.e., the variation pattern of colonization rate with root order and the differences).

Response: We agree, and we actually found a lower mycorrhizal colonization rate at the fifth root order, but colonization rate in the fourth root order was similar to the third root order (the following figure). We have now added this Figure to the Supplementary Information (Figure S4).

4. In addition, considering the species specificity, if the colonization rate of the 4/5-order roots in your study is similar to that of the absorptive roots, does this mean that in the tree population you studied, the higher-level root orders should be the transport roots? In other words, without looking at the anatomy, the method of directly dividing the absorptive vs transport roots by root order has great limitations, which need to be pointed out in the paper.

Response: We agree, and we are aware that mycorrhizal colonization rate is highly dependent on species, and the transition from absorptive to transport roots might again be different across different species and might occur gradually. Of course, the best way to do this differentiation would be to look into the root anatomical changes of each species. However, as we have shown in the above figure, the cortex-to-stele ratio as an anatomical trait showed a decreasing trend across root order, with a steeper decrease in the last two orders (the fourth and fifth). We have now added some explanations why mycorrhizal colonization rate results might not have shown differences across the root functional types in the Introduction (Lines 91-92) and Discussion (Lines 269-287) sections.

5. Trees with high growth rates are theoretically expected to correspond to traits related to a fast-slow strategy dimension. However, in this case, the traits are more closely related to mycorrhizal traits, especially mycorrhizal colonization rates. Additionally, both composite and individual leaf traits are positively correlated with average basal area increment (which reflects tree growth). These two results seem somewhat counterintuitive, so you may need to explore these issues further in your discussion.

Response: We are a bit uncertain what the reviewer means with this comment. Maybe the reviewer is thinking that individual (referring to the single trait-growth relationship) and composite (putting all traits together in PCA) leaf traits both increased the growth. As for leaves, the bivariate regression showed that leaf mass per area (LMA) was positively significantly and leaf dry matter content (LDMC) was marginally positively correlated with average basal area increment (Figure S3kl). Similarly, the PCA scores-growth relationships showed that species with higher LMA and LDMC had

a higher growth (Figures 3c,4c). This means that species with a conservative strategy (high LDMC and LMA) has a higher growth, which contrasts with the expectations of the leaf economic spectrum concept.

In addition, we found the same directionality in the bivariate root trait-growth relationships and also the PCA scores-growth relationships (Figures 3, S3). For example, in absorptive root traits, the bivariate regressions showed that tree growth was negatively correlated with root diameter (RD), cortex to stele ratio (C:S) and mycorrhizal colonization rates (MCR) (Figure S3a,d,e). We found similar results with the same directionality in the PCA scores-growth relationships, where tree species with thicker roots and lower MCR grew better (Figures 3a,4a). Overall, in absorptive roots, traits related to mycorrhizal (RD, C:S and MCR) are more significantly negatively related to tree growth.

Considering this comment, we have revised and expanded the discussion section, providing more mechanistic insights about how above and belowground traits decoupled (Line 341-350).

6. The clarification on whether coordination exists between absorptive roots and transport roots is still insufficient. This is evident in two aspects: first, the presentation of data, and second, the discussion in your manuscript. A possible approach to address this could be to supplement your analysis by including both absorptive and transport root traits in a PCA to observe the trait distribution patterns.

Response: This is a great point; thanks for bringing up this point. We have now addressed this point in the Discussion section and provided a PCA combining both absorptive and transport root traits (Figure S2, Table S2). In short, this analysis showed that the same root traits on absorptive roots were quite strongly related to the same traits on transport roots (e.g. Figure. S2: RTD of both root types were positively loaded to PC2, MCR and C:S to PC1 and SRL negatively to PC2). There was one big difference: root diameter, which mostly related to PC3 for transport roots, and to PC1 for absorptive roots (Table S2).

7. Lines 300-302: The slow strategy, based on leaf traits, corresponds to higher tree growth rates, suggesting a decoupling between roots and leaves in explaining tree growth. This raises several unclear issues: the alignment and directionality of the strategy dimension represented by traits in explaining tree growth, and the inconsistency among different organs. These issues need to be better clarified and discussed in more detail.

Response: Many thanks for this helpful point. Considering your comment, we have now added sentences in this context in the Discussion section (Lines 341-350).

8. Lines 155-156: Would it be more appropriate to focus on absorptive roots in this hypothesis, considering that absorptive roots, like leaves, are organs responsible for resource acquisition?

Response: We have now revised the hypothesis as suggested (Lines 161-163).

9. Line 195 (Figure 2d) and Line 314 (Figure 1f). There are many similar figure citation errors in the full text, which will reduce the readability of the paper. Please check it throughout the text.

Response: We have now carefully double-checked the Figures citations throughout the text.

Reviewers' comments:

Reviewer #1:

I am glad to say that all my concerns have been adequately addressed. The data analysis and the figures have been revised according to my suggestions. From my perspective, the manuscript can now be accepted. Congratulations to the authors for the great work!

Response: We acknowledge the helpful comments and suggestions provided by the reviewer. We're glad to know that our revisions and responses fulfilled the reviewer comments. Thank you for recommending our research for publication.

Reviewer #2:

The paper is actually improved and all my raised comments have been addressed. In my opinion, it is now ready for publication.

Colours are not explained in Figure 3

Response: We thank you for your thoughtful assessment of our manuscript and for your suggestions to improve it. We are glad the reviewer believes our manuscript should be acceptable for publication following revisions.

We have now explained colors in Figures 2 (lines 756-757) and 3 (lines 771-772) as follows:

Different colors (line or point) represent different tree species. See Table 1 for tree species name abbreviations.

We also added a header to the legend to clarify this point.

Reviewer #3:

The authors have already addressed most of the issues I raised, but there are still a few issues I would like to provide feedback to the authors.

Response: We thank the reviewer for the comprehensive and thorough assessment. We tried to address all of these comments with great care, and provide point by point responses below.

1. Regarding the fourth question, specifically lines 270-285, I believe the discussion here could be more concise. Additionally, it might be more appropriate to adopt a tone that frames it as "future research needs to...".

Response: Thank you for the suggestion; we have now added the following sentence to the corresponding discussion as suggested.

Furthermore, due to the unique anatomical structures of different species and the resulting variation in root functions across root orders, further research is needed to precisely distinguish between the absorptive and transport functions of each tree species based on their anatomical changes (lines 296-299).

2. Regarding the fifth question, your answer "the bivariate regressions showed that tree growth was negatively correlated with root diameter (RD), cortex to stele ratio (C:S) and mycorrhizal colonization rates (MCR) (Figure S3a,d,e). We found similar results with the same directionality in the PCA scores-growth relationships, where tree species with thicker roots and lower MCR grew better (Figures 3a,4a)."

Please carefully verify the accuracy and consistency of the descriptions and figures here, and everywhere else in the main text except in this reply.

Response: We have now mentioned the consistency between single and/or multiple traits-tree growth relationships in both results (lines 219-226) and discussion (already mentioned in the previous version; lines 322-332) sections.

3. Regarding the sixth question, specifically lines 241-244, please carefully check the results in Fig. 3 and Fig. S2, and clarify which of your results are similar to those of previous studies. Does your Fig. 3b support the claim that species with larger root diameters were highly related to mycorrhizal association?

Response: Thank you for highlighting this. We have now revised the text as follows:

Despite significant differences between absorptive and transport root traits (Figure 1), we found that, contrary to our first hypothesis (H1), coordination within absorptive and transport root traits was quite similar to each other (Figures 3a,b, S2), with the exception of RD, which was decoupled from MCR and C:S in transport roots (Figures 2,3b). Our findings show that species with higher root diameter were highly related to mycorrhizal association, but this was true only for absorptive roots, similar to a previously published study (lines 247-252).